# Plan To Predict: Learning an Uncertainty-Foreseeing Model for Model-Based Reinforcement Learning

**Zifan Wu**
School of Computer Science and Engineering
Sun Yat-sen University
Guangzhou, China
`wuzf5@mail2.sysu.edu.cn`

**Chao Yu**[*]
School of Computer Science and Engineering
Sun Yat-sen University
Guangzhou, China
`yuchao3@mail.sysu.edu.cn`

**Chen Chen**
Noah's Ark Lab
Huawei
Beijing, China
`cclvr@163.com`

**Jianye Hao**
Noah's Ark Lab
Huawei
Beijing, China
`haojianye@huawei.com`

**Hankz Hankui Zhuo**
School of Computer Science and Engineering
Sun Yat-sen University
Guangzhou, China
`zhuohank@mail.sysu.edu.cn`

## Abstract

In Model-based Reinforcement Learning (MBRL), model learning is critical since an inaccurate model can bias policy learning via generating misleading samples. However, learning an accurate model can be difficult since the policy is continually updated and the induced distribution over visited states used for model learning shifts accordingly. Prior methods alleviate this issue by quantifying the uncertainty of model-generated samples. However, these methods only quantify the uncertainty passively after the samples were generated, rather than foreseeing the uncertainty before model trajectories fall into those highly uncertain regions. The resulting low-quality samples can induce unstable learning targets and hinder the optimization of the policy. Moreover, while being learned to minimize one-step prediction errors, the model is generally used to predict for multiple steps, leading to a mismatch between the objectives of model learning and model usage. To this end, we propose *Plan To Predict* (P2P), an MBRL framework that treats the model rollout process as a sequential decision making problem by reversely considering the model as a decision maker and the current policy as the dynamics. In this way, the model can quickly adapt to the current policy and foresee the multi-step future uncertainty when generating trajectories. Theoretically, we show that the performance of P2P can be guaranteed by approximately optimizing a lower bound of the true environment return. Empirical results demonstrate that P2P achieves state-of-the-art performance on several challenging benchmark tasks.

## 1   Introduction

Through learning an approximate dynamics of the environment and then using it to assist policy learning, Model-based Reinforcement Learning (MBRL) methods [Moerland et al., 2020] have been shown to be much more sample-efficient than model-free methods both theoretically and empirically [Nagabandi et al., 2018, Luo et al., 2019, Janner et al., 2019]. The learning of the model is critical for MBRL, since an inaccurate model can bias policy learning through generating misleading pseudo samples, which is also referred to as the model bias issue [Deisenroth and Rasmussen, 2011].

---

[*]corresponding author

36th Conference on Neural Information Processing Systems (NeurIPS 2022).

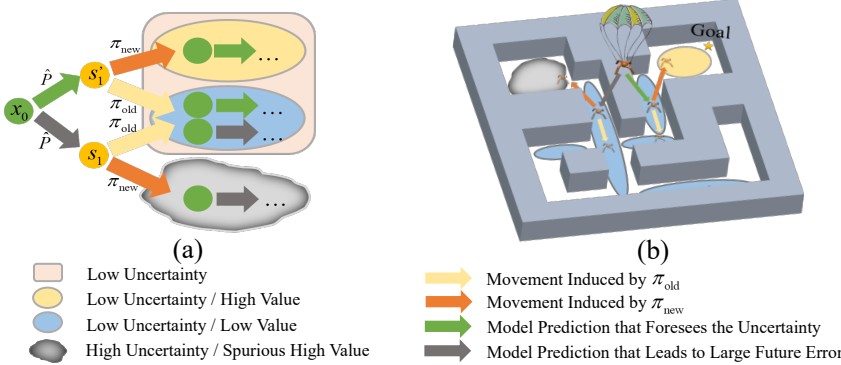

Figure 1: (a) Intuition for an uncertainty-foreseeing model; (b) A specific maze task modified from the D4RL benchmark [Fu et al., 2020], where an ant falls from the sky and randomly lands on the two sides of a wall. The ant aims to reach the goal using shortest time (assuming that the goal cannot be seen from the sky).

However, learning an accurate model can be difficult, since the update of the policy continually shifts the distribution over visited states, and due to lack of enough training data for the modified distribution, the model can suffer from overfitting and become inaccurate in regions where the updated policy is likely to visit. This issue is further aggravated when using expressive models like deep neural networks, which are inherently prone to overfitting.

Prior methods mitigate this issue by applying uncertainty quantification techniques, typically by using a bootstrapped model ensemble [Kurutach et al., 2018, Chua et al., 2018, Janner et al., 2019]. However, due to the lack of explicit considerations of the aforementioned policy shift issue, these methods can only quantify the uncertainty passively after the samples were generated, and thus fail to prevent the trajectories induced by the current policy from running into regions with high uncertainty. This incapability to avoid uncertain regions can result in low quality of the generated samples, which provides unstable learning targets and thus may hinder the learning of the policy. Moreover, the model in the existing methods is usually used to predict for multiple steps, while the model learning objective is commonly to minimize the one-step prediction errors. In other words, there exists an objective mismatch [Lambert et al., 2020] between model learning and model usage, which means that the policy optimization generally expects accurate multi-step trajectories (i.e., model usage), yet the model is only estimated to consider the accuracy of immediate predictions (i.e., model learning).

To gain a conceptual understanding of the above two motivations, consider the scenario in Figure 1(a): 1) The approximate model $\hat{P}$ of the true environment dynamics $P$ predicts $\hat{P}(s_1|x_0) > 0, \hat{P}(s_1'|x_0) > 0$ and $\hat{P}(s|x_0) = 0, \forall s \notin \{s_1, s_1'\}$ at the state-action pair $x_0$; and 2) The policy is updated from $\pi_{\text{old}}$ to $\pi_{\text{new}}$ to exploit the regions with high values predicted by the value function in the model, and the resulting distribution shift exposes the model's inaccuracy at $s_1$ to the current policy, leading to large expected prediction error in the next step, i.e., $\sum_{a'} \pi_{\text{new}}(a'|s_1)D\big(\hat{P}(\cdot|s_1, a'), P(\cdot|s_1, a')\big) >> \sum_{a'} \pi_{\text{new}}(a'|s_1')D\big(\hat{P}(\cdot|s_1', a'), P(\cdot|s_1', a')\big)$, where $D(\cdot, \cdot)$ is some distance metrics and the symbol $>>$ means significantly greater than. Now starting from $x_0$, under the updated policy $\pi_{\text{new}}$, predicting $s_1$ is likely to direct the trajectory into highly uncertain regions with spurious high values (e.g., the grey region in Figure 1(b)), and cause considerably larger error in the subsequent trajectory generation than that of predicting $s_1'$. In contrast, if the model can foresee this uncertainty through explicit considerations of the current policy, then it can learn to plan for future predictions and thus jump to $s_1'$ more often to prevent the trajectory from falling into uncertain regions (e.g., the ant falls into the right hand side of the wall in Figure 1(b)). Therefore, to generate trajectories that have small accumulation errors, the model should be aware of the update of the policy and change its predicting "strategy" by taking into account the long-term effect of immediate predictions.

Inspired by the above intuition, we propose a new framework for MBRL, named *Plan To Predict* (P2P). Our core idea is to treat the model rollout process as a sequential decision making problem and learn the model by the corresponding algorithms, such as reinforcement learning and model predictive control. During the learning of the model, the current policy is fixed and serves as the environment dynamics, while at the same time the model becomes a decision maker and is penalized via the

prediction errors at each step. Through actively interacting with the current policy and minimizing the accumulative prediction errors on the resulting state distribution, the model can quickly adapt to the state distribution shift by considering the long-term effect of the immediate predictions, thus foreseeing the uncertainty of regions that model trajectories are likely to fall into. Such foresight of uncertainty enables the model to plan to predict for multiple steps, which improves the quality of the pseudo samples generated by the model and eventually benefits policy learning. Note that the typical treatment of minimizing one-step prediction errors regardless of the current policy is in fact a greedy optimization process, and thus can be viewed as a special case of P2P. Some prior works, such as [Nagabandi et al., 2018, Luo et al., 2019], adopt similar multi-step prediction loss to learn the model, yet these works simply compute the loss on trajectories collected by previous policies and can still suffer from severe distribution shift due to the lack of considerations of the policy update.

Theoretically, we prove that P2P approximately optimizes a lower bound of the true environment return. Different from prior works [Luo et al., 2019, Janner et al., 2019], our theoretical analysis does not scale up the step-wise prediction errors to their maximum over timesteps when considering the impact of model error on the return approximation, thus resulting in a tighter bound of the true return and also justifying the objective mismatch issue. Optimizing this bound not only provides a stronger guarantee for policy improvement, but also enables the model to quickly adapt to the current policy. Empirical results demonstrate that P2P outperforms the existing state-of-the-art methods on several challenging benchmark tasks. It also deserves to note that, due to the capability to avoid model error, P2P may be particularly suited for offline learning tasks, where incremental data collection is not allowed and thus the model error is harder to be corrected than in online settings, inducing an irreversible bias on policy learning. Therefore, we also provide an evaluation of P2P in offline learning settings in the experiments, and present an interpretative visualization on how the learning objective of P2P affects the model predictions.

## 2    Preliminaries and Notations

A discrete time Markov desision process (MDP) $\mathcal{M}$ is defined by tuple $(\mathcal{S}, \mathcal{A}, R, P, \rho_0, \gamma)$, where $\mathcal{S}$ is the state space, $\mathcal{A}$ the action space, $R(s, a)$ a scalar reward function of $s \in \mathcal{S}$ and $a \in \mathcal{A}$, $P(s'|s, a) \in [0, 1]$ the transition dynamics which is assumed to be unknown, $\gamma \in [0, 1)$ the discount factor and $\rho_0$ the initial state distribution. A stochastic policy $\pi(a|s)$ maps states to a probability distribution over the action space. The expected return of $\pi$ defined by $J(\pi) := \mathbb{E}_{\pi, P} \left[ \sum_{t=0}^{\infty} \gamma^t R(s_t, a_t) \right]$ is the expectation of the sum of discounted rewards induced by $\pi$. Following the general setting of reinforcement learning, the goal is to search for an optimal policy that maximizes the expected return from the environment, i.e., $\arg \max_\pi J(\pi)$.

MBRL methods learn a model $\hat{P}$ to approximate the unknown dynamics $P$, and then use this model to aid policy learning. While the model can be utilized in various ways, this paper focuses on the usage of generating pseudo samples to enrich the dataset, which is one of the most common usages of the model. The expected return of $\pi$ predicted by the model $\hat{P}$ is denoted as $J^{\hat{P}}(\pi) := \mathbb{E}_{\pi, \hat{P}} \left[ \sum_{t=0}^{\infty} \gamma^t R(s_t, a_t) \right]$. In our analysis, the reward function is assumed to be available, [2] and the policy obtained in the last iteration is denoted as the data-collecting policy $\pi_D$,

## 3    The Plan To Predict (P2P) Framework

In this section, we present a detailed description of our proposed algorithmic framework for MBRL, i.e., *Plan To Predict* (P2P). In Section 3.1, the model rollout process is reformulated as a sequential decision making problem using the notion of MDP and then a meta-algorithm of the P2P framework is provided as a generic solution to this reformulated problem. In Section 3.2, theoretical results are provided to guarantee the performance of P2P. In Section 3.3, two practical algorithms are developed under the P2P framework by handling some practical issues.

---

[2]Note that this is a commonly used assumption since the sample complexity of learning the reward function with supervised learning is a lower order term compared to the one of learning the transition model [Gheshlaghi Azar et al., 2013].

---
**Algorithm 1** Meta-Algorithm of the P2P Framework
---
1: Initialize the policy $\pi$ and the approximate model $\hat{P}$;
2: Initialize empty dataset $\mathcal{D}$;
3: **for** each epoch **do**
4:     Collect data with $\pi$ in real environment: $\mathcal{D} \leftarrow \mathcal{D} \cup \{(s_i, a_i, r_i, s'_i)\}_i$;
5:     Optimize $\hat{P}$ for multi-step prediction under the current policy $\pi$: $\hat{P} \leftarrow \arg\max_{\hat{P}'} J^\pi(\hat{P}')$
6:     Optimize $\pi$ under $\hat{P}$: $\pi \leftarrow \arg\max_{\pi'} \hat{J}^{\hat{P}}(\pi')$
7: **end for**
---

## 3.1 The Overall Framework

We now define the *model MDP* to reformulate the model rollout problem and outline our overall algorithm design.

**Definition 1.** *The model MDP $\mathcal{M}_\pi$ is defined by tuple $(S^m, A^m, R^m, P^m, \gamma, \rho_0^m)$, where at timestep $t$, $s_t^m := (s_t, a_t) \in S^m$, $a_t^m := s_{t+1} \in A^m$, $R^m(s_t^m, a_t^m) := -\left(\hat{P}(s_{t+1}|s_t, a_t) - P(s_{t+1}|s_t, a_t)\right)^2$, $P^m(s_{t+1}^m|s_t^m, a_t^m) := \pi(a_{t+1}|s_{t+1})\hat{P}(s_{t+1}|s_t, a_t), \gamma$ is the discount factor , $\rho_0^m := \pi(a_0|s_0)\rho_0$, and $\rho_0$ is the initial state distribution of the true environment.*

In the model learning process of P2P, the roles played by the policy $\pi$ and the model $\hat{P}$ are reversed: $\pi$ is fixed to serve as the environment dynamics and $\hat{P}$, now as the decision maker, interacts with $\pi$ and is updated to minimize the accumulative prediction error along the generated trajectories. Similar to the optimization of the policy, the objective of this learning process can be written as $\arg\max_{\hat{P}} J^\pi(\hat{P})$ where $J^\pi(\hat{P}) := \mathbb{E}[\sum_t \gamma^t R^m(s_t^m, a_t^m)]$. The overall algorithmic framework of P2P is depicted in Algorithm 1, where the main difference compared to most existing MBRL methods lies in Line 5, i.e., the learning of the model. Specifically, this process can be achieved in a way similar to policy learning, i.e., generating samples by actively interacting with the policy (now viewed as the background environment), and optimizing the expected return $J^\pi(\hat{P})$ accordingly. This optimization can in principle be done by any approaches solving sequential decision making problems. As two representatives of these approaches, *Model Predictive Control* (MPC) [Camacho and Alba, 2013] and *Reinforcement Learning* (RL) [Sutton and Barto, 2018] are both applied in our practical version of P2P, which are detailed in Section 3.3. In addition, one of the biggest gaps between this theoretical construction of the *model MDP* and the practical implementation may lie in the derivation of $R^m$. According to Definition 1, the true environment dynamics is required to compute this reward function, while in practice the true dynamics is not accessible in the model learning phase. Therefore, depending on which underlying approach (i.e., MPC or RL) is chosen for model optimization, we propose two different alternative plans to approximate $R^m$ in Section 3.3.

## 3.2 Theoretical Results

In general, MBRL methods learn an approximate model of the real world dynamics and learn the policy by maximizing the expected return predicted by the model. The better this pseudo return approximates the true environment return, the stronger guarantee the model can provide for policy improvement. Thus, a critical problem in theory is how to bound the gap between the expected return of the model and the true environment, i.e., $\left|J^{\hat{P}}(\pi) - J(\pi)\right|$. Suppose that this gap can be upper bounded by $C$, and the policy is updated from $\pi_D$ to $\pi$ with the expected model return improved by $2C$, then the policy improvement in terms of the true environment can be guaranteed:

$$J(\pi) - J(\pi_D) \geq \left(J^{\hat{P}}(\pi) - C\right) - \left(J^{\hat{P}}(\pi_D) + C\right)$$
$$\geq J^{\hat{P}}(\pi) - J^{\hat{P}}(\pi_D) - 2C \geq 0.$$

Our main theoretical result is the following theorem:

**Theorem 1.** *The gap between the expected return of the model and the environment is bounded as:*

$$\left|J(\pi) - J^{\hat{P}}(\pi)\right| \leq \frac{2R_{max}}{(1-\gamma)^2}\left((2-\gamma)\epsilon_\pi + (1-\gamma)\sum_{t=1}^{\infty}\gamma^t\epsilon_t^m\right), \tag{1}$$

where $\epsilon_\pi := \max_s D_{TV}(\pi_D(\cdot|s)\|\pi(\cdot|s))$ denotes the policy distribution shift, $\epsilon_t^m := \mathbb{E}_{s\sim\hat{P}_{t-1}(s,a;\pi)}\big[D_{TV}(\hat{P}(\cdot|s,a)\|P(\cdot|s,a))\big]$ denotes the upper bound of one-step model prediction error at timestep $t$ of the model rollout trajectory, $D_{TV}(p\|q)$ refers to the total variation between distribution $p$ and $q$, $R_{max} := \max_{s,a} R(s,a)$, and $\hat{P}_{t-1}(s,a;\pi)$ denotes the state-action distribution at $t$ under $\hat{P}$ and $\pi$.

*Proof.* Please refer to Appendix A. □

Transforming the inequality in Eq. (1) results in $J(\pi) \geq J^{\hat{P}}(\pi) - 2C(\hat{P},\pi)$, where $C(\hat{P},\pi)$ denotes the right hand side in Eq. (1). Thus by iteratively applying the update rule in Eq. (2), the above lower bound of the policy performance can be monotonically improved:

$$\pi, \hat{P} \leftarrow \arg\max_{\pi,\hat{P}} J^{\hat{P}}(\pi) - 2C(\hat{P},\pi). \qquad (2)$$

This update rule is ideal and usually impractical since it involves an exhaustive search in the state and action space to compute $C$, and requires full-horizon rollouts in the model for estimating the accumulative model errors. Thus, similar to how algorithms like TRPO [Schulman et al., 2015] approximate their theoretically monotonic version, P2P approximates this update rule by maximizing the expected model return while keeping the accumulative model error small. Specifically, via replacing the computationally expensive $D_{TV}$ with an expectation over $\hat{P}$ in the definition of $\epsilon_t^m$, it can be easily derived that maximizing $J^\pi(\hat{P})$ defined in Section 3.1 is equivalent to minimizing $\sum_t \gamma^t \epsilon_t^m$. As for the policy shift term $\epsilon_\pi$, though the bound suggests that this term may also be constrained, we found empirically that it is sufficient to only control the model error. This may be explained by the relatively small scale of policy shift with respect to model error, as observed in [Janner et al., 2019].

It is worth noting that the accumulative error here is not simply summing up the multi-step errors on trajectories collected by any other policies as in [Nagabandi et al., 2018, Luo et al., 2019], because the definition of the step-wise model error (i.e., $\epsilon_t^m$) in Theorem 1 involves a policy-dependent expectation over the state space, indicating that this error is also affected by the update of the policy. To enable quick adaptation to this policy update, P2P minimizes the accumulative error on the state distribution induced by an active interaction between the model and the current policy, and thus better approximates the accumulative error defined in Theorem 1.

Different from most of prior work [Luo et al., 2019, Janner et al., 2019], in Theorem 1 the step-wise model prediction errors are not scaled up to their maximum over timesteps (i.e., $\max_t \epsilon_t^m$), leading to a tighter error bound. As a result, this bound not only provides stronger guarantee for policy improvement, but also justifies the objective mismatch issue between the model learning and model usage from a theoretical perspective: Scaling up these error terms by such a potentially large magnitude yields a loose bound which leads to algorithms aiming at minimizing the upper bound of individual one-step prediction losses. However, to obtain better policy improvement, the tighter bound in Eq. (1) implies that a more fine-grained model learning process is desired, which considers consecutive rollout steps and minimizes the accumulative prediction error, i.e., $\sum_t \gamma^t \epsilon_t^m$. Using the notion of *model MDP* defined in Section 3.1, the model induced by minimizing one-step prediction errors can be intuitively interpreted as a greedy decision maker, which is often considered shortsighted and may easily lead to severe suboptimality.

### 3.3 Practical Implementation

In this subsection, we instantiate Algorithm 1 by specifying explicit approaches to learn the model and providing solutions to some practical issues. Detailed pseudo codes can be found in Appendix E.

**P2P-MPC:** Since in our problem formulation the "dynamics" of the model rollout process, i.e., the current policy $\pi$, is accessible, one of the simplest yet effective ways to learn the model can be the model predictive control (MPC) [Camacho and Alba, 2013], which utilizes the dynamics to plan and optimize for a sequence of actions. Given the state $s_t^m$ at step $t$, the MPC controller first optimizes the sequence of actions $a_{t:t+H}^m$ over a finite horizon $H$, and then employs the first action of the optimal action sequence, i.e., $a_{H,t}^m := \arg\max_{a_{t:t+H}^m} \mathbb{E}_{\hat{P}} \sum_{t'=t}^{t+H-1} R^m(s_{t'}^m, a_{t'}^m)$. Computing the

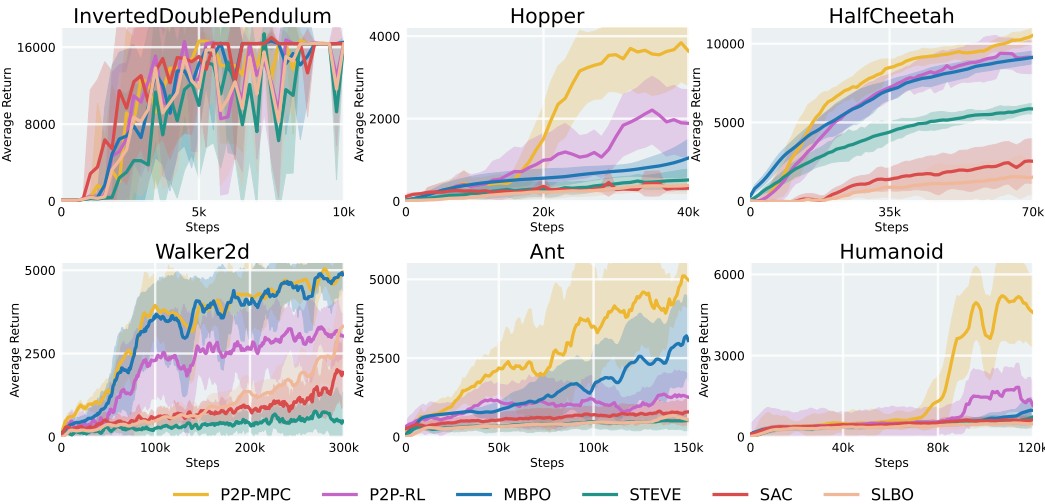

Figure 2: Comparisons against baselines on MuJoCo. Solid curves represent the mean of runs over 5 different random seeds, and shaded regions correspond to standard deviation among these runs.

exact $\arg\max$ is computationally challenging due to the dimension of the state and action spaces, so we adopt the random-sampling shooting method [Rao, 2009] which generates random action sequences, executes them respectively, and chooses the one with highest return predicted by the dynamics. Besides, the reward $R^m$, which cannot be directly computed by definition, is approximated here using a neural network and trained on the environment dataset. Specifically, during each training iteration, P2P-MPC first trains the model via traditional one-step prediction loss, and then trains the $\hat{R}^m$ network by taking transitions sampled from the environment dataset as inputs, and the minus prediction errors on these transitions as labels. The prediction error of an environment transition $(s, a, r, s')$ is computed via $\|\hat{s}' - s'\| + \|\hat{r} - r\|$, where $\hat{s}', \hat{r}$ are sampled from $\hat{P}(\cdot, \cdot|s, a)$.

**P2P-RL:** It is well-known that though effective, RL methods often suffer from high sample complexity. Hence, the model is trained on the environment dataset instead of the samples generated by the interaction of the model and policy. However, there are two issues to be addressed: 1) As the "environment dynamics", the policy is highly non-stationary due to its continuing update; and 2) The decision maker which generates the dataset is the true environment rather than our model. The first issue can be addressed by updating the next state in each sampled transitions by applying the current policy, i.e., $s^m_{t+1} \leftarrow (s_{t+1}, \pi(s_{t+1}))$. As for the second issue, since the model-generated samples are not used in model learning, it can be seen approximately as an offline learning problem and thus can be addressed by well-developed offline RL methods [Fujimoto et al., 2019, Levine et al., 2020]. Due to the effectiveness and the implementational simplicity, we adopt SAC [Haarnoja et al., 2018] with behavior cloning as the underlying learning algorithm, which bears similarities to the TD3+BC approach proposed by [Fujimoto and Gu, 2021]. Besides, an approach called DualDICE [Nachum et al., 2019] is also applied to correct the estimation of the state distribution. Finally, since the samples come from the true environment, the reward $R^m$ can be simply approximated by re-predicting the dynamics in the sampled transitions and computing the prediction errors for the current model. Specifically, P2P-RL trains the model on the environment dataset and treats the model learning process as an offline RL problem, where the true dynamics becomes the "decision maker" of the environment dataset in our problem formulation. Thus, regarding a transition $(s, a, r, s')$, $R^m$ can be directly approximated by computing $-\|\hat{s}' - s'\| - \|\hat{r} - r\|$, where $\hat{s}', \hat{r} \sim \hat{P}(\cdot, \cdot|s, a)$.

## 4 Experiments

In this section, we first evaluate the empirical performance of P2P on a set of MuJoCo continuous control tasks [Todorov et al., 2012] in Section 4.1, and then present an quantitative analysis for the model error in Section 4.2. The ablation study provided in Section 4.3 verifies the necessity of an active policy-model interaction when computing the multi-step prediction loss. Besides, a key

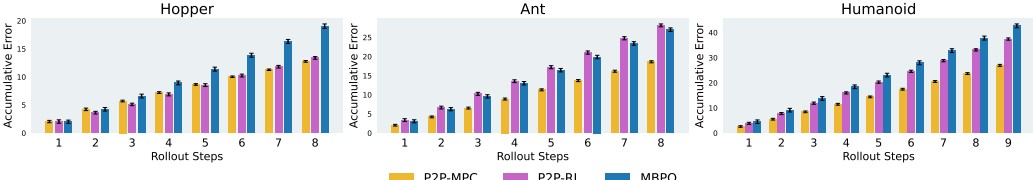

Figure 3: Quantitative analysis of the accumulative model errors. While the true environment dynamics is not accessible during learning, the errors are approximated via supervised learning on the environment dataset.

parameter of P2P-MPC is also studied in Section 4.3. Finally, to gain a deeper understanding of how P2P work in offline settings, a visualization of the model's rollout trajectories is shown in Section 4.4.

### 4.1 Comparisons with Baselines

We compare P2P [3] against some state-of-the-art model-free and model-based methods. The model-free baseline is SAC [Haarnoja et al., 2018], which is the model-free counterpart of P2P. The model-based baselines include MBPO [Janner et al., 2019], which uses short-horizon rollouts branched from the state distribution of previous policies, SLBO [Luo et al., 2019], which enjoys theoretical performance guarantee and uses model rollouts from the initial state distribution, and STEVE [Buckman et al., 2018], which also generates short rollouts but uses model data for estimating target values instead of policy learning. Note that our framework only focuses on model learning and can be employed as a flexible plug-in component by most MBRL algorithms that generate pseudo samples to assist policy learning. In our experiments, we plug the model learning process of P2P into MBPO since it is a widely accepted strong baseline in MBRL. The implementational details and the hyperparameter settings of P2P, as well as the environment settings, are all described in Appendix B.

As shown in Figure 2, P2P-MPC significantly outperforms the baselines on several tasks, especially on Hopper, Humanoid and Ant, which are generally considered to be the most challenging MuJoCo tasks. In Walker2d, the best rollout length is one step in our implementation and MBPO, thus resulting in similar performance between these two methods. Compared to P2P-MPC, the performance of P2P-RL is less stable. Through an investigation on the model learning process (see Appendix C), we find that P2P-RL sometimes struggles to balance the loss of behavior cloning and RL, leading to the difficulty in hyperparameter tuning and the instability of learning network parameters, while the optimization of P2P-MPC is non-parametric and thus does not suffer from this issue.

### 4.2 Model Error Analysis

The theoretical result presented in Section 3.2 implies that to guarantee better policy improvement, the model should learn to minimize the accumulative errors along the trajectories induced by the current policy and the model. Since P2P is designed based on this principle, the empirical results in Figure 2 verify the theoretical result in terms of the final performance. To further verify the effectiveness of this algorithmic design, we study the accumulative prediction error on three tasks that require rollouts with relative long horizons, i.e., Hopper, Ant and Humanoid. The results shown in Figure 3 together with Figure 2 strongly support our theoretical claims by demonstrating that the algorithm inducing less accumulative error achieves better performance. Besides, we can also observe that in Hopper the model errors of P2P-MPC in the first few steps are slightly larger than the other two methods. Nevertheless, as the rollout trajectory goes longer, the accumulative error induced by P2P-MPC grows significantly slower than the other methods. This further agrees with the intuition mentioned in Figure 1 by showing that P2P is able to trade the one-step model error for accumulative model error by considering the long-term effect of the immediate prediction.

### 4.3 Ablation Study

**Multi-Step Prediction Loss** In Section 3.2, we emphasize that the loss of P2P is essentially different from the multi-step prediction loss used in prior work, since P2P computes the multi-step loss on the state distribution induced from an active interaction between the model and the current

---

[3]Code available at `https://github.com/ZifanWu/Plan-to-Predict`.

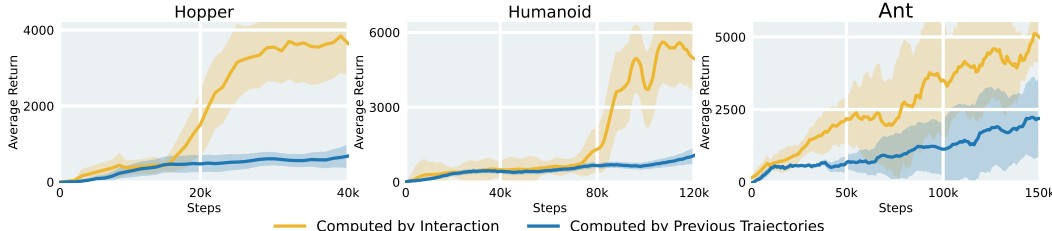

Figure 4: Results comparing performances induced by two ways of computing the multi-step prediction loss. The yellow curves represent the performance of P2P-MPC, which minimizes the multi-step loss on the trajectories generated by active interactions between the model and the current policy. The blue curves show the results of an ablation version of MBPO where the original one-step loss is replaced by a multi-step loss computed over the trajectories sampled from the environment dataset. The lengths of these trajectories are set to the same in this comparison.

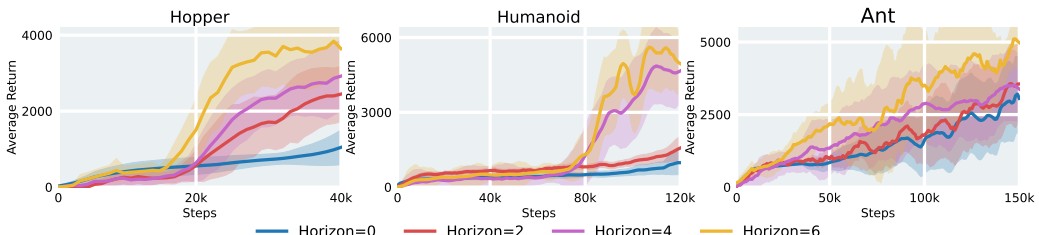

Figure 5: Results comparing performances induced by different planning horizons of the model in P2P-MPC.

policy, and on the contrary, prior work computes this loss on the trajectories collected by previous policies. The empirical results in Figure 4 compare the performances induced by these two kind of losses and highlight the importance of the interactive model learning process in P2P.

**Planning Horizon** Different from most RL methods, MPC methods can assign an explicit horizon for planning, hence P2P-MPC is able to set the planning horizon of the model to be exactly the same with the rollout horizon. This difference may partially explain the superior performance of P2P-MPC against P2P-RL. Nevertheless, longer planning horizon incurs higher computational cost, especially in tasks with high-dimensional state spaces. To trade-off the performance and the computational cost, our solution is to simply limit the planning horizon, and the empirical results shown in Figure 5 suggests that a horizon of 6 is satisfactory for the tasks experimented.

## 4.4 An Interpretative Experiment

To gain an intuitive understanding of how the learning objective of P2P affects the model predictions, here we present a visualization of the rollout trajectories in maze2d-medium, an offline learning task in the D4RL benchmark [Fu et al., 2020]. As shown in Figure 6, the environment is slightly modified by lengthening the wall at the middle of the maze and fixing the goal to the top right corner. Besides, the number of offline samples that involve the grey region are partially discarded to increase this region's uncertainty. Figure 6 shows the top four trajectories that are most frequently generated from a fixed starting point using our methods and MOPO [Yu et al., 2020], a well-known offline model-based RL method that uses one-step error for model learning. The visualization clearly demonstrates the different rollout preferences of the two kinds of models: the model of P2P-MPC tends to generate trajectories heading to the regions with low uncertainty, while the model of MOPO fails to prevent the trajectories from being absorbed into the highly uncertain region. The resulting accumulative model errors are quantified in Figure 7, and as shown in Table 1, the final performance of the induced policy of P2P-MPC is significantly better than MOPO. Note that in online settings, the model error is possible to be corrected by the newly collected data from the environment, while in offline settings this incremental data-collecting process is not allowed. Thus, the model error can have a

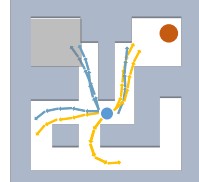

Figure 6: Visualization of models' rollout trajectories. The blue trajectories are generated by MOPO, while the yellow ones are generated by P2P-MPC.

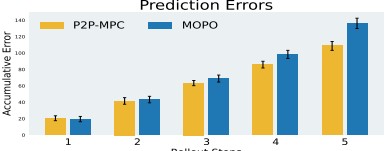

Figure 7: Accumulative prediction error along the rollout trajectories on the modified maze2d-medium task.

Table 1: Comparison with MOPO on two maze tasks.

| Environment | P2P-MPC | MOPO |
|---|---|---|
| maze2d-medium (modified) | 230.2 $\pm$ 39.5 | 174.2 $\pm$ 32.4 |
| maze2d-large-diverse | 8.1 $\pm$ 1.1 | 0.8 $\pm$ 0.0 |

more detrimental effect on policy learning, which may severely degrade the asymptotic performance as evidenced in Table 1. Additional results on a much harder task (i.e., maze2d-large-v1) are also provided in Table 1 to further verifies the effectiveness of P2P-MPC.

## 5 Related Work

MBRL aims at improving the sample efficiency of model-free methods while maintaining high asymptotic performance. The research of MBRL can be roughly divided into two lines: the model usage and the model learning. The most common usage of the model is generating pseudo samples to enrich the data buffer, so as to reduce the interaction with the environment and thus accelerate policy learning [Sutton, 1990, 1991, Deisenroth et al., 2013, Kalweit and Boedecker, 2017, Luo et al., 2019, Janner et al., 2019, Pan et al., 2020]. The main challenge of this kind of usage is the compounding model error of long-term predictions, which is also known as the model bias issue [Deisenroth and Rasmussen, 2011]. A number of approaches are proposed to address this issue, varying from using Dyna-like short-horizon rollouts [Sutton, 1991, Janner et al., 2019], interpolating between real and pseudo samples [Kalweit and Boedecker, 2017], learning an uncertainty-aware probabilistic model ensemble [Chua et al., 2018], to masking the rollouts according to the uncertainty estimation [Pan et al., 2020]. Falling into this category of model usage, the focus of our work is to improve the quality of model-generated trajectories by reducing the accumulative model error along these trajectories. Through an active interaction between the model and the current policy, P2P reinforces a quick model adaptation to the update of policy, so as to foresee and thus avoid the uncertainty in the future trajectory during rollout. Hence, one of the major differences between P2P and prior work can be interpreted as: not just to passively estimate the uncertainty after encountering it, but to actively avoid it before this encounter.

It is worth noting that there are also methods based on the optimism in face of uncertainty, which try to exploit the uncertainty of the model for sufficient exploration in the real environment, such as [Curi et al., 2021]. If the model's promise of large return turns out to be false, executing the policy trained by such a model will hopefully collect real-world counterfactual data to the previously erroneous model. However, this correcting process itself requires additional samples, which may run counter to the principle of developing model-based methods. Furthermore, exploring the erroneous part of the model may not only provides the agent with false information about the real environment, but also be prone to the model exploitation issue which severely hurts the asymptotic performance of the policy. Therefore, how to carefully exploit the model uncertainty remains an open question.

In the line of model learning, the most commonly used objective is to simply minimize each one-step prediction error for transitions available in the environment dataset [Kurutach et al., 2018, Chua et al., 2018, Janner et al., 2019]. However, since the data distribution of the environment dataset varies with the update of policy, it is unlikely to learn an accurate model for all the transitions in complex environments. Thus, when the model is used to predict for multiple steps, there exists an objective mismatch [Lambert et al., 2020] between model usage and model learning. To address this issue, we propose to minimize the accumulative prediction error under the state distribution of the current policy. The most relevant work to ours includes [Nagabandi et al., 2018, Luo et al., 2019], where a similar multi-step prediction loss is employed for model learning. The essential difference between their multi-step loss and ours is clarified in Section 3.1 and 3.2. Generative models [Kingma and Welling, 2014] have also been utilized to perform multi-step prediction [Mishra et al., 2017, Ha and Schmidhuber, 2018]. However, the need for a highly exploratory dataset limits their application to relatively simple and low-dimensional tasks. Other learning objectives improve traditional model learning from angles that are less similar to ours, including goal-aware model

learning that predicts how far the current state is from the goal state [Nair et al., 2020] and decision-aware model learning that aims to better aids policy learning by re-weighting the model error on different samples [Farahmand et al., 2017, Farahmand, 2018, D'Oro et al., 2020].

In recent years, a series of work has attempted to learn the model by treating the model rollout process as a sequential decision-making problem. While these work shares some similarity with our work, the targeted problems and the proposed solutions are not exactly the same as P2P. Shang et al. [2019] propose an environment reconstruction method which models the influence of the hidden confounder on the environment by treating the platform, the user and the confounder as three agents interacting with each other. They focus on the offline setting (i.e., RL-based recommendation) and simultaneously train the model and the policy using a multi-agent imitation learning method. Xu et al. [2020] treat the model as a dual agent and analyze the error bounds of the model. They propose to train the model using imitation learning methods. Chen et al. [2022] also consider multi-step model error, yet they mainly focus on handling counterfactual data queried by adversarial policies. Unlike the above two work [Xu et al., 2020, Chen et al., 2022] that both focus solely on model learning, P2P aims at proposing a general MBRL framework and attempts to adapt the model to the continuously updating policy quickly.

## 6    Conclusion and Future Work

In this paper, we propose an MBRL framework named Plan To Predict (P2P), which novelly treats the model rollout process as a sequential decision making problem. In this problem reformulation, the roles played by the model and the policy are reversed, where the model becomes a decision maker and the policy instead serves as the outer environment. Through an active interaction between the model and the current policy, the model can minimize the multi-step accumulative errors on the induced trajectories and thus quickly adapt to the state distribution of the current policy. We provide theoretical guarantee for P2P, which also clarifies the difference of our multi-step loss from prior methods and justifies the mismatch of objectives between model learning and model usage. Empirical results on several challenging benchmark tasks verify the effectiveness of P2P. Moreover, observing that the capability to avoid model error makes P2P promising for offline settings, we also provide preliminary results on some offline learning tasks. While this is not the major focus of this paper, studying how P2P can be applied in offline tasks more properly is left as an important future work.

## Acknowledgments and Disclosure of Funding

The work is supported by the National Natural Science Foundation of China under Grant 62076259. The authors would like to thank Qian Lin, Zongkai Liu and Xuejing Zheng for the useful discussions, and thank Hanlin Yang for the help in drawing Figure 1 (b). Finally, the reviewers and metareviewer are highly appreciated for their constructive feedback on the paper.

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
