## A    Mathematical Proofs

Here we provide proofs for Theorem 1 in the main paper.

**Theorem.** *The gap between the expected return of the model and the environment is bounded as:*

$$\left| J(\pi) - J^{\hat{P}}(\pi) \right| \leq \frac{2R_{max}}{(1-\gamma)^2} \left( (2-\gamma)\epsilon_\pi + (1-\gamma) \sum_{t=1}^{\infty} \gamma^t \epsilon_t^m \right), \tag{1}$$

*where $\epsilon_\pi := \max_s D_{TV}(\pi_D(\cdot|s) \| \pi(\cdot|s))$ denotes the policy distribution shift, $\epsilon_t^m := \mathbb{E}_{s \sim \hat{P}_{t-1}(s,a;\pi)} \left[ D_{TV}(\hat{P}(\cdot|s,a) \| P(\cdot|s,a)) \right]$ denotes the upper bound of one-step model prediction error at timestep t of the model rollout trajectory, $D_{TV}(p\|q)$ refers to the total variation between distribution p and q, $R_{max} := \max_{s,a} R(s,a)$, and $\hat{P}_{t-1}(s,a;\pi)$ denotes the state-action distribution at t under $\hat{P}$ and $\pi$.*

*Proof.*

$$\left| J^{\hat{P}}(\pi) - J(\pi_D) \right| = \left| \sum_{t=0}^{\infty} \gamma^t \sum_{s,a} \left( \hat{P}_t^\pi(s,a) - P_t^{\pi_D}(s,a) \right) R(s,a) \right|$$

$$\leq R_{max} \sum_{t=0}^{\infty} \gamma^t \sum_{s,a} \left| \hat{P}_t^\pi(s,a) - P_t^{\pi_D}(s,a) \right|$$

$$\leq 2R_{max} \sum_{t=0}^{\infty} \gamma^t D_{TV} \left( \hat{P}_t^\pi(s,a) \| P_t^{\pi_D}(s,a) \right). \tag{2}$$

Applying Lemma 1, we have:

$$D_{TV} \left( \hat{P}_t^\pi(s,a) \| P_t^{\pi_D}(s,a) \right) \leq D_{TV} \left( \hat{P}_t^\pi(s) \| P_t^{\pi_D}(s) \right) + \epsilon_\pi. \tag{3}$$

Similar to the proof of Lemma B.2 in [Janner et al., 2019], we have:

$$\left| \hat{P}_t^\pi(s) - P_t^{\pi_D}(s) \right| = \left| \sum_{s'} \hat{P}^\pi(s_t = s \mid s') \hat{P}_{t-1}^\pi(s') - P^{\pi_D}(s_t = s \mid s') P_{t-1}^{\pi_D}(s') \right|$$

$$\leq \sum_{s'} \left| \hat{P}^\pi(s_t = s \mid s') \hat{P}_{t-1}^\pi(s') - P^{\pi_D}(s_t = s \mid s') P_{t-1}^{\pi_D}(s') \right|$$

$$= \sum_{s'} \left| \hat{P}^\pi(s \mid s') \hat{P}_{t-1}^\pi(s') - P^{\pi_D}(s \mid s') \hat{P}_{t-1}^\pi(s') + P^{\pi_D}(s \mid s') \hat{P}_{t-1}^\pi(s') - P^{\pi_D}(s \mid s') P_{t-1}^{\pi_D}(s') \right|$$

$$\leq \sum_{s'} \hat{P}_{t-1}^\pi(s') \left| \hat{P}^\pi(s \mid s') - P^{\pi_D}(s \mid s') \right| + P^{\pi_D}(s \mid s') \left| \hat{P}_{t-1}^\pi(s') - P_{t-1}^{\pi_D}(s') \right|$$

$$= \mathbb{E}_{s' \sim \hat{P}_{t-1}^\pi} \left[ \left| \hat{P}^\pi(s \mid s') - P^{\pi_D}(s \mid s') \right| \right] + \sum_{s'} P^{\pi_D}(s \mid s') \left| \hat{P}_{t-1}^\pi(s') - P_{t-1}^{\pi_D}(s') \right|. \tag{4}$$

Then, the term $D_{TV} \left( \hat{P}_t^\pi(s) \| P_t^{\pi_D}(s) \right)$ can be bounded by:

$$D_{TV}\left(\hat{P}_t^\pi(s)\|P_t^{\pi_D}(s)\right) = \frac{1}{2}\sum_s \left|\hat{P}_t^\pi(s) - P_t^{\pi_D}(s)\right|$$

$$= \frac{1}{2}\mathbb{E}_{s'\sim\hat{P}_{t-1}^\pi}\left[\sum_s \left|\hat{P}^\pi(s\mid s') - P^{\pi_D}(s\mid s')\right|\right] + \frac{1}{2}D_{TV}\left(\hat{P}_{t-1}^\pi(s')\|P_{t-1}^{\pi_D}(s')\right)$$

$$= \frac{1}{2}\sum_{t'=1}^t \mathbb{E}_{s'\sim\hat{P}_{t'-1}^\pi}\left[\sum_s \left|\hat{P}^\pi(s|s') - P^{\pi_D}(s|s')\right|\right]$$

$$= \frac{1}{2}\sum_{t'=1}^t \mathbb{E}_{s'\sim\hat{P}_{t'-1}^\pi}\left[\sum_s \left|\sum_a \hat{P}^\pi(s,a|s') - P^{\pi_D}(s,a|s')\right|\right]$$

$$\leq \frac{1}{2}\sum_{t'=1}^t \mathbb{E}_{s'\sim\hat{P}_{t'-1}^\pi}\left[\sum_{s,a} \left|\hat{P}^\pi(s,a|s') - P^{\pi_D}(s,a|s')\right|\right]$$

$$= \sum_{t'=1}^t \mathbb{E}_{s'\sim\hat{P}_{t'-1}^\pi} D_{TV}\left(\hat{P}^\pi(s,a|s')\|P^{\pi_D}(s,a|s')\right). \tag{5}$$

12 Again applying Lemma 1, we have:

$$D_{TV}\left(\hat{P}^\pi(s,a|s')\|P^{\pi_D}(s,a|s')\right) \leq \epsilon_\pi + \mathbb{E}_{a\sim\pi} D_{TV}\left(\hat{P}(s|s',a)\|P(s|s',a)\right). \tag{6}$$

13 Plugging Eq. (6) in Eq. (5) we have:

$$D_{TV}\left(\hat{P}_t^\pi(s)\|P_t^{\pi_D}(s)\right) \leq \sum_{t'=1}^t \epsilon_\pi + \epsilon_{t'}^m = t\epsilon_\pi + \sum_{t'=1}^t \epsilon_{t'}^m. \tag{7}$$

14 Plugging Eq. (7) in Eq. (3) we have:

$$D_{TV}\left(\hat{P}_t^\pi(s,a)\|P_t^{\pi_D}(s,a)\right) \leq t\epsilon_\pi + \sum_{t'=1}^t \epsilon_{t'}^m + \epsilon_\pi = (t+1)\epsilon_\pi + \sum_{t'=1}^t \epsilon_{t'}^m. \tag{8}$$

15 Plugging Eq. (8) in Eq. (2) we have:

$$\left|J^{\hat{P}}(\pi) - J(\pi_D)\right| \leq 2R_{max}\sum_{t=0}^\infty \gamma^t \left((t+1)\epsilon_\pi + \sum_{t'=1}^t \epsilon_{t'}^m\right)$$

$$= 2R_{max}\left(\frac{\epsilon_\pi}{(1-\gamma)^2} + \frac{1}{(1-\gamma)}\sum_{t=1}^\infty \gamma^t \epsilon_t^m\right)$$

$$= \frac{2R_{max}}{(1-\gamma)^2}\left(\epsilon_\pi + (1-\gamma)\sum_{t=1}^\infty \gamma^t \epsilon_t^m\right). \tag{9}$$

16 Therefore, the result in Eq. (1) can be derived:

$$\left|J(\pi) - J^{\hat{P}}(\pi)\right| \leq |J(\pi) - J(\pi_D)| + \left|J(\pi_D) - J^{\hat{P}}(\pi)\right|$$

$$\leq \frac{2R_{max}\epsilon_\pi}{1-\gamma} + \frac{2R_{max}}{(1-\gamma)^2}\left(\epsilon_\pi + (1-\gamma)\sum_{t=1}^\infty \gamma^t \epsilon_t^m\right)$$

$$= \frac{2R_{max}}{(1-\gamma)^2}\left((2-\gamma)\epsilon_\pi + (1-\gamma)\sum_{t=1}^\infty \gamma^t \epsilon_t^m\right).$$

17 $\qquad\qquad\qquad\qquad\qquad\qquad\qquad\qquad\qquad\qquad\qquad\qquad\qquad\qquad\qquad\qquad\qquad\qquad\square$

**Lemma 1.** *(TVD of joint distribution) Suppose that we have two distributions $P_1(x, y) = P_1(x)P_1(y|x)$ and $P_2(x, y) = P_2(x)P_2(y|x)$. We can bound the total variance difference of the joint as:*

$$D_{TV}(P_1(x, y)\|P_2(x, y)) \leq D_{TV}(P_1(x)\|P_2(x)) + \mathbb{E}_{x \sim P_1}[D_{TV}(P_1(y|x)\|P_2(y|x))].$$

Lemma 1 is proved in the MBPO paper, so we only provide the result here.

## B   Experimental Details

We describe some implementation details and hyperparameter settings below.

### B.1   Implementation and Hyperparameter Settings

Our implementation is overall based on MBPO. The algorithm for policy learning, the actor-critic network architecture, the boostrapped model ensemble technique and other details are all the same with those in MBPO. The only modified part is the model learning process which is the focus of the main paper. In P2P-MPC, the candidate action sequences are generated in a parallel manner to accelerate this process, at the cost of some extra memory cost. The number of candidate sequences is set to 4 for Hopper, Ant and Humanoid, and 6 for HalfCheetah. In P2P-RL, we adopt the normalization technique used in TD3+BC for the state and RL loss, and the hyperparameter $\alpha$ is set to 2. The model is trained 50 times for InvertedDoublePendulum, Hopper and HalfCheetah, and 20 times for rest of the tasks. To learn the $\zeta$ and $\nu$ network in DualDICE, we first train them for 1e5 times at the 5-th epoch and then train them for 2 times every time before policy learning. The learning rate of these two networks are both set to 1e-4 and the batch size is set to 1024.

### B.2   Environment Settings

All the environments remain the same with the original version of the tasks, except for the Inverted-DoublePendulum task where some additional noises are added to the states, which are set to Gaussian noises with mean 0 and standard deviation 10.

## C   Analysis of the Model Learning Process of MPC-RL

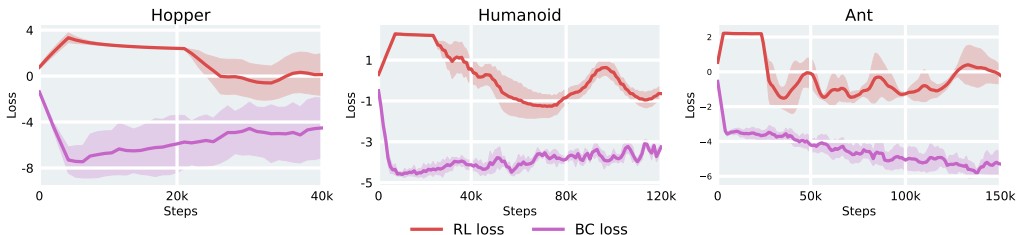

Figure 1: Quantitative analysis of the two kinds of losses in the model learning process of MPC-RL. "RL loss" means the loss of the reinforcement learning objective, and "BC loss" means the loss of the behavior cloning objective.

As shown in Section 4.1, the performance of P2P-RL is generally worse than P2P-MPC. An investigation on the model learning process is shown in Figure 1, implying that the cause of this suboptimality may be the difficulty in balancing the loss of behavior cloning and RL.

## D   Extended Experiment for the Case When the Goal is in an Uncertain Region

Serving as an extended experiment for the maze experiment in Section 4.4, we investigate the case when the goal is in an uncertain region in the online setting. For the convenience of im-

plementation, here the term "uncertainty" is equated with the epistemic uncertainty [Chua et al., 2018], which can be quantified as the amount of relevant real-world data. Therefore, a region with more data is considered to have lower uncertainty. Since in pure online settings the uncertainty of regions is hard to control during the training iterations, we first pretrain the model with an offline dataset and then switch to online training. The goal is allocated to the grey region in Figure 6 where the relevant offline samples are partially discarded. The percent of discarded samples is set to 25%, 50%, 75% and 100% respectively and the results are shown in Table 1.

| | **25%** | **50%** | **75%** | **100%** |
|---|---|---|---|---|
| P2P-MPC | $148.9 \pm 35.9$ | $75.4 \pm 31.6$ | $51.7 \pm 29.8$ | $43.2 \pm 25.1$ |
| MBPO | $116.2 \pm 35.6$ | $61.1 \pm 34.8$ | $47.5 \pm 35.1$ | $44.7 \pm 30.2$ |

Table 1: Results on the scenario where goal is in an uncertain region.

As the degree of uncertainty increases, the performances of both methods degrade rapidly, but P2P-MPC still outperforms MBPO in all these cases except for the 100% case, where P2P-MPC achieves slightly worse performance in average but better stability with lower standard deviation. To give a possible explanation of these results, it is worth noting that 1) P2P does not directly intervene the learning of policy or value function, but only improves the accuracy of the generated samples. As a result, the value function can still predict high value for uncertain regions and thus encourage the policy to explore them in the real environment; and 2) in contrast, even if the goal is in a region of high uncertainty and the model does not prevent the policy from exploring this region in the model, the value function can still predict low value of this region due to the lack of relevant data and thus mislead the learning of policy.

# E Pseudo Code

The detailed descriptions of P2P-MPC and P2P-RL are respectively provided in Algorithm 1 and Algorithm 2.

---

**Algorithm 1** P2P-MPC

---

1: Initialize policy $\pi$, predictive model $\hat{P}$, model-error predictor $\hat{R}^m$, environment dataset $\mathcal{D}_e$ and model dataset $\mathcal{D}_m$.
2: **for** $N$ epochs **do**
3:     Train model $\hat{P}$ on $\mathcal{D}_e$ via one-step prediction loss;
4:     Train $\hat{R}^m$ on $\mathcal{D}_e$;
5:     **for** $E$ steps **do**
6:         Take action in environment according to $\pi$, and store the new transition to $\mathcal{D}_e$;
7:         **for** $M$ model rollouts **do**
8:             Sample initial states $s \sim \mathcal{D}_e$ uniformly for rollout trajectories;
9:             **for** $k$ rollout steps **do**
10:                 Take action $a$ according to $\pi$ and the current state $s$ in the model;
11:                 Initialize $s_0^m = (s, a)$ and perform $L$ parallelized rollouts for $H$ steps;
12:                 Compute $\sum_{t=0}^{H-1} \hat{R}^m(s_{t,j}^m, a_{t,j}^m)$ for each rollout trajectory $j$, denoted as $r_j^m, j \in \{1, 2, ..., L\}$, and take $(s', r) = a_{0, \arg\max_j r_j^m}^m$;
13:                 Store $(s, a, r, s')$ to $\mathcal{D}_m$ and then Let $s = s'$;
14:             **end for**
15:         **end for**
16:         **for** $G$ gradient updates **do**
17:             Update $\pi$ using data sampled from $\mathcal{D}_m$;
18:         **end for**
19:     **end for**
20: **end for**

---

---

**Algorithm 2** P2P-RL

---

1: Initialize policy $\pi$, predictive model $\hat{P}$, environment dataset $\mathcal{D}_e$, model dataset $\mathcal{D}_m$, $\zeta$ and $\nu$ for using of DualDICE.
2: **for** $N$ epochs **do**
3:     Train the $\zeta$ and $\nu$ network according to the following objective derived by [Nachum et al., 2019]: $\mathbb{E}_{\mathcal{D}_e}[(\nu(s_t^m, a_t^m) - \gamma\nu(s_{t+1}^m, a_{t+1}^m))\zeta(s_t^m, a_t^m) - f^*(\zeta(s_t^m, a_t^m)) - (1-\gamma)\nu(s_0^m, a_0^m)]$, where $f^*(x) := \frac{2}{3}|x|^{\frac{2}{3}}$ and $s_{t+1}^m$ is updated from $(s_{t+1}, \pi_D(s_{t+1}))$ to $(s_{t+1}, \pi(s_{t+1}))$. Note that here $\pi_D$ means the data-collecting policy and $\pi$ the current policy;
4:     Train $\hat{P}$ by optimizing $\mathbb{E}_{s_t^m, r_t^m \sim \mathcal{D}_e, a^m \sim \hat{P}(\cdot|s_t^m)}[\zeta(s_t^m, a^m)(\log \hat{P}(a^m|s_t^m) - Q^m(s_t^m, a^m)) + r_t^m]$, where the first term is the SAC [Haarnoja et al., 2018] loss and the second is the behavior cloning loss. Note that $Q^m(s^m, a^m)$ is the action-value function of $\hat{P}$ and is trained by the same objective as the one in SAC;
5:     **for** $E$ steps **do**
6:         Take action $a_t$ in environment according to $\pi$ and the current state $s_t$, then obtain the next state $s_{t+1}$ and the reward $r_{t+1}$;
7:         Reorder the transition: $(s_t, a_t, r_{t+1}, s_{t+1}, \hat{r}_{t+1}, \hat{s}_{t+1}) \rightarrow (s_t^m, a_t^m, r_{t+1}^m, s_{t+1}^m)$, where $s_t^m = (s_t, a_t), a_t^m = (\hat{r}_{t+1}, \hat{s}_{t+1} - s_{t+1}), r_{t+1}^m = -\|\hat{s}_{t+1} - s_{t+1}\| - \|\hat{r}_{t+1} - r_{t+1}\|$, and $s_{t+1}^m = (s_{t+1}, a_{t+1})$ if $t + 1 \leq E$ else $(s_E, \pi(s_E))$; Store $(s_t^m, a_t^m, r_{t+1}^m, s_{t+1}^m)$ into $\mathcal{D}_e$;
8:         **for** $M$ model rollouts **do**
9:             Sample initial states uniformly from $\mathcal{D}_e$;
10:             Perform $k$-step model rollouts starting from these states, using $\pi$ and $\hat{P}$; Add the generated samples to $\mathcal{D}_m$;
11:         **end for**
12:         **for** $G$ gradient updates **do**
13:             Update $\pi$ using data sampled from $\mathcal{D}_m$;
14:         **end for**
15:     **end for**
16: **end for**

---