# OpenReview forum: "Plan To Predict: Learning an Uncertainty-Foreseeing Model For Model-Based Reinforcement Learning"
_NeurIPS.cc/2022/Conference — NeurIPS 2022 Accept_

### Official Review · Reviewer_N1PQ · 2022-06-19

**Rating:** 7
**Confidence:** 5
**Soundness:** 4 excellent
**Presentation:** 4 excellent
**Contribution:** 4 excellent

**Summary:**

Model based reinforcement learning promises to reduce sample complexity in reinforcement learning. MBRL is responsible for some of the major breakthroughs in AI, zuch as AlphaGo Zero, and AlphaFold. MBRL may suffer from model inaccuracy, especially when the learning of the model, which is typically doen with 1 step lookahead, is separate from model usage, which is typically done with many step lookahead. The paper suggests a solution to this problem, using a meta-approach, to approach the model learning as an MDP problem itself.
The paper offers both a theoretical analysis, and convincing experimental evidence of strongly better performance than some of the best model based, and some of the best model free approaches. This a convincing contribution.

**Questions:**

- can you compare to more other MBRL approaches
- can you run on larger problems?

**Limitations:**

few limitations are mentioned in the paper. The authors are invited to provide more
There are no ethical issues to address

**Strengths And Weaknesses:**

Strengths
- clear writing, clear problem statement
- clear contribution
- good theoretical analysis
- convincing experimental evidence
- important new idea, that performs substantially better. important contribution

Weaknesses
- few

---

> ### Author Response · Authors · 2022-08-02
> **Response to Reviewer N1PQ**
>
> We thank the reviewer for all these valuable comments. We provide point-by-point responses below.
>
> **Q1: "Can you compare to more other MBRL approaches?"**
>
> **A1:** We have added one more baseline, i.e., SLBO [1], in the evaluation of our revised version (Section 4.1, Page 7, Lines 226-227).
>
>
>
> **Q2: "Can you run on larger problems?"**
>
> **A2:** In this work, we conducted experiments on MuJoCo and D4RL, which are the widely used benchmarks in the existing MBRL research. Extending our work to larger problems is left as an important future work.
>
>
>
> [1] Luo et al. Algorithmic framework for model-based deep reinforcement learning with theoretical guarantees. 2019.

---

### Official Review · Reviewer_xhVo · 2022-07-05

**Rating:** 7
**Confidence:** 4
**Soundness:** 3 good
**Presentation:** 2 fair
**Contribution:** 4 excellent

**Summary:**

The paper proposes an improved method for learning models for model-based reinforcement learning, by recasting the supervised regression approach commonly used as a control problem.

**Questions:**

- Figure 1: while nicely designed, I do not think I fully understand what the figure represents. It has a lot of details and an unfamiliar setup. Please simplify and clarify.
- Line 42: Since the term "objective mismatch" is used, I would encourage the authors to cite previous work by Lambert et al. and others.
- Justification: It is unclear to me why exactly the model should prevent the policy from going into high uncertainty regions. First, in most model based methods, several trajectories would be computed (explicitly with random shooting, or implicitly, by repeatedly querying a probabilistic model). These should to some extend counteract the overestimation, as the model would be less controllable in this region. Furthermore, executing the next policy on the real environment will hopefully provide counterfactual data to the previously erroneous model.
- How does predicting $R^m$ actually differ between this approach and MBPO? How is the expectation with regard to the current policy realized?
- Line 162: I do not fully understand what is meant by this sentence? Is the difference in the data used for the update?
- Line 190: The $R^m$ is trained as a neural network? How exactly? This seems crucial for making the method work, but is treated as a side comment.

I think a lot of my confusion stems from the last question: how is the reward/loss for the mode actually computed. I think a minimal change that would alleviate a lot of my concerns would be to add a more clear, step by step explanation of the algorithm to the top of Section 3, instead of relegating implementation details to later? Since the setup is fairly different from other model learning approaches, a more "tutorial style" of writing would immensely benefit my (and hopefully others) understanding.

**Limitations:**

The method discusses an optimization method and therefore does not clearly require a discussion of societal impact.

**Strengths And Weaknesses:**

The following will be very brief and most of my comments will be under "Questions", because I have serious doubts that I actually understood the method the authors are proposing. This is not due to lack of familiarity with the field, I have done research and published in model-based reinforcement learning. I will point out my problems in the next section.

Given the very nice results and the overall introduction, I am very favorable to recommend acceptance. This is not currently reflected in the score however, due to the questions I have about the paper. Therefore I would ask the authors to go through them and hopefully improve the presentation (or convince me that I overlooked crucial details of course :) ), and to not be discouraged by the low score! I think the ideas warrant a presentation at NeurIPS, but I think that the paper can be heavily improved to make it accessible and give it a chance to shine in the community.

---

> ### Author Response · Authors · 2022-08-02
> **Response to Reviewer xhVo**
>
> We thank the reviewer for all these valuable comments. Point-by-point responses are provided below.
>
> **Q1: Clarification of details and setup in Figure 1.**
>
> **A1:** Figure 1(a) is a conceptual model that illustrates the our motivation, and the setup can be described as follows: Given an arbitrary state-action pair $x_0$, the model has two options of prediction, namely $s_1$ and $s_1'$. Under the old policy $\pi_{\text{old}}$, both options will lead the trajectory to enter regions with low value, hence $\pi_{\text{old}}$ is updated to $\pi_{\text{new}}$ to explore regions with potential high value.  Under the current policy $\pi_{\text{new}}$, predicting $s_1$ will result in a subsequent trajectory with significantly higher accumulative error than that of predicting $s_1'$.
>
> Figure 1(b) is an informal instance of Figure 1(a), where $x_0$ corresponds to the ant falling from the sky (executing action like adjusting the belt of the parachute), $s_1$ and $s_1'$ respectively corresponds to landing on the left/right side of the wall, and the arrows as well as the colored regions have the same meaning with the ones in Figure 1(a).
>
> We have simplified Figure 1 in the introduction of our revised version (Page 2).
>
>
>
> **Q2: Cite additional previous work for using the term "objective mismatch".**
>
> **A2:** Thanks for reminding us of this. We have cited the mentioned work in our revision (Page 2, Line 42).
>
>
>
> **Q3: Justification of "why exactly the model should prevent the policy from going into high uncertainty regions".**
>
> **A3:** First, from the theoretical perspective, preventing the policy from going into highly uncertain regions can reduce the accumulative model error, and thus guarantee a tighter performance lower bound and better policy improvement according to Theorem 1.
>
> Second, as mentioned by the reviewer, the overestimation of unfamiliar states can to some extent be counteracted by utilizing large batch of parallelized trajectories during model rollout. However, it is often not clear that, to what extent and how large the batch size can counteract this overestimation, and there is no guarantee that the expectation of these parallelized samples can well approximate the true value. From this perspective, the potential risk of overestimation may still be high if the policy visits the uncertain regions in the model frequently. That is why the performance becomes better if the model can foresee this risk and avoid it in advance. This is also the main motivation of our work.
>
> Finally, as mentioned by the reviewer, the erroneous part of the model can be hopefully corrected by counterfactual data in the online setting. However, this correcting process itself requires additional samples, which may run counter to the principle of developing model-based methods. Furthermore, exploring the erroneous part of the model may not only provides the agent with false information about the real environment, but also be prone to the *model exploitation* issue which severely hurts the asymptotic performance of the policy.
>
> **Q4: "How does predicting** $R^m$ **actually differ between this approach and MBPO? How is the expectation with regard to the current policy realized?"**
>
> **A4:** Roughly speaking, $R^m$ takes a transition tuple as input and return the model error on this transition. In this regard, the key difference between P2P and MBPO can be described as: MBPO optimizes $E_{s_0,a_0,s_1}[R^m]$ where $s_0\sim p_{\pi_{\text{old}}}, a_0\sim\pi_{\text{old}}(\cdot|s_0),s_1\sim \hat{P}(\cdot|s_0,a_0)$, and P2P optimizes $E_{s_{0:T+1},a_{0:T}}[\sum\gamma^tR^m_t]$ where $s_0\sim p_{\pi_{\text{old}}}$ and $a_t\sim\pi_{\text{new}}(\cdot|s_t),s_{t+1}\sim \hat{P}(\cdot|s_t,a_t)$ for $t\in\{0, \ldots,T\}$. Since $s_1'\sim P(\cdot|s_0, a_0)$ can be approximated by sampling from the environment dataset, MBPO updates $\hat{P}$ by directly minimizing $\\|s_1-s_1' \\|$. In contrast, $(s_t,a_t)$ may be not available in the environment dataset since $s_t$ is predicted by the model and $a_t$ is obtained from the new policy that have not interacted so much with the true environment. In other words, in practice P2P needs to approximate $R^m$ to predict the model accuracy on unseen inputs. We will clarify the specific approach for this approximation in **A6**.
>
> The expected return with regard to the current policy is approximated through finite trajectory samples generated by active interactions between the fixed current policy and the model.
>
>
>
> **Q5: The meaning of Line 162.**
>
> **A5:** As discussed in **A4**, in theory P2P optimizes $E_{s_{0:T+1},a_{0:T}}[\sum\gamma^tR^m_t]$ where $s_0\sim p_{\pi_{\text{old}}}$ and $a_t\sim\pi_{\text{new}}(\cdot|s_t),s_{t+1}\sim \hat{P}(\cdot|s_t,a_t)$ for $t\in\{0, \ldots,T\}$. To approximate the above expectation, the model needs to interact with the current policy $\pi_{\text{new}}$ and optimize over the induced data.

---

> > ### Author Response · Authors · 2022-08-02
> > **Response to Reviewer xhVo**
> >
> > **Q6: "Line 190: The $R^m$ is trained as a neural network? How exactly?"**
> >
> > **A6:** In practice, during each training iteration, P2P-MPC first trains the model via traditional one-step prediction loss, and then trains the $\hat{R}^m$ network by taking transitions sampled from the environment dataset as input, and the prediction errors on these transitions as label. The prediction error of an environment transition $(s, a, r, s')$ are computed via $\\|\hat{s}'-s'\\|+\\|\hat{r}-r\\|$, where $\hat{s}', \hat{r}$ are sampled from $\hat{P}(\cdot, \cdot|s, a)$. The above details are added to Appendix B.3 in our revised version.
> >
> > As discussed in **A4**, to computed the expected return w.r.t. the current policy, $\hat{R}^m$ may need to predict the model accuracy on unseen transitions, and this requirement for generalization is why we choose to use a neural network. Intuitively, $\hat{R}^m$ can be seen as an indicator that tells the model where its "weakness" lies in.
> >
> > The above clarification has been added to Appendix B.3 (Page 3, Lines 38-48) in the revised version.

---

> > > ### Comment · Reviewer_xhVo · 2022-08-07
> > > **Thanks**
> > >
> > > Dear authors, thanks for humoring my problems and expanding the paper! In its current forms, the details and benefits of the formulation are much more clear to me! I will be raising my score to "Accept".
> > > I am still confused about Figure 1 and I would encourage you to expand the description in the final paper, similar to the text you wrote in the rebuttal comment. I would potentially suggest to have only the left version of the figure but separated into old and new policy side, with highlighted differences? This is just a suggestion for improvement in the camera ready though.

---

> > > > ### Author Response · Authors · 2022-08-08
> > > > **Response to Reviewer xhVo**
> > > >
> > > > We thank the reviewer for the valuable suggestions and for updating the score. We will further expand the description of Figure 1  in our future revision to improve the readability.

---

> ### Author Response · Authors · 2022-08-04
> **Response to Reviewer xhVo**
>
> Thank you for the suggestion of using a more "tutorial style" of writing. We have rewrote Section 3.1 to give a more specific explanation of the algorithm. Please refer to Lines 124-136 in our revision.
>
> Please let us know if there are still confusions.

---

### Official Review · Reviewer_q2Vv · 2022-07-11

**Rating:** 6
**Confidence:** 3
**Soundness:** 2 fair
**Presentation:** 3 good
**Contribution:** 2 fair

**Summary:**

The paper proposes a model-based reinforcement learning algorithm that alternates between model-learning with the policy fixed, and policy improvement with the model fixed. The key insight is that the model-learning phase is treated as a Markov decision process (MDP) with the estimated model as the "policy" to be learned (with the actual policy fixed), where the reward is the negative mismatch between the estimated model and the actual system transition dynamics. The paper claims that encapsulating model-learning with a fixed policy as a "reversed" MDP encourages model-learning to more quickly adapt to policy updates and produce a model that is more accurate over multiple transition steps.

**Questions:**

Based on the discussion above in "Strengths and Weaknesses", I have the following questions and requests for clarification:

- Is the reward function R(s,a) assumed to be known? If not, where and how do you learn R(s,a) in Algorithm 1?

- For P2P-MPC, on lines 190-191 the authors state "the reward Rm, which cannot be directly computed by definition, is approximated here using a neural network and trained on the environment dataset." For P2P-RL, on the lines 205-207 the authors state "since the samples come from the true environment, the reward Rm can be simply approximated by re-predicting the dynamics in the sampled transitions and computing the prediction errors for the current model-policy". Both statements lack significant detail and yet seem to be critical to model learning since they quantify the difference between the model P_hat and the true transition dynamics P. Please expand on these points.

- Please expand on your evaluations of the model error induced by P2P and multi-step loss function baselines, particularly in Figure 4. As it stands, the lack of details severely hampers reproducibility.

- The authors promote that their proposed method P2P learns by "actively interacting with the current policy" (line 66) and has the "capability to avoid model error" (line 84). Distinctly, the authors' ant-in-a-maze example scenario has the goal in a region of low model uncertainty. However, it is certainly possibly to encounter a scenario where the goal is in a region of high model uncertainty, thereby requiring further exploration in the online setting. Could P2P underperform or fail in such scenarios? Further experiments would be welcome here.


**Limitations:**

There is no substantial discussion of limitations in this work. There is some superficial comparison of P2P-MPC and P2P-RL at the end of Section 4.1 and in Appendix C, but little insight or discussion is provided. I have raised an issue regarding exploration-exploitation in the review sections above that may be useful for the authors to consider.

The authors have adequately addressed the societal impact of their work.


**Strengths And Weaknesses:**

*** ORIGINALITY ***

The key contribution of the paper is the treatment of the unknown transition model as the "decision-maker" when paired with a fixed policy during model learning. The novelty of this idea is clearly established by the authors in their comparisons to prior work. The paper proposes the "meta-algorithm" Algorithm 1, which requires the user to choose from existing algorithms to specify the model and policy learning phases. Overall, the proposed novelty can be viewed as a specification on the objective for the model learning phase. However, the empirical improvements offered by this key insight are clearly demonstrated in Section 4, so there is strong value in it as a contribution.

*** QUALITY ***

The paper is technically sound for the most part, and the methods used seem appropriate. However, it is unclear if the authors learn the reward function in Algorithm 1 at all -- there does not seem to be a description anywhere in the main paper if R(s,a) is known or must be learned. In addition, in Figure 4 of Section 4, the results cannot be interpreted because the authors have not provided a clear description of what these plots show. Indeed, the caption only vaguely states that these results compare "performances induced by two ways of computing the multi-step prediction loss". Since the authors repeatedly tout the benefit of their approach over prior approaches using multi-step objectives, this part of Section 4.3 should be expanded and improved.

*** CLARITY ***

The paper is clearly written for the most part. Some notation in Theorem 1 is undefined in the main body of the paper and in the appendix (e.g., R_max, D_TV). The use of the term "value" throughout the paper should be clarified (e.g., does this refer to a value function?). The use of the terms "policy-dynamics" for the policy pi and "model-policy" for the dynamics model P_hat beginning on line 180 is confusing. It would probably be clearer to just refer to them as the policy and model, respectively, and ensure it is clear to the reader when either the policy-learning or the model-learning phase is being discussed.

*** SIGNIFICANCE ***

The key idea in treating dynamics learning with a fixed policy as an MDP with the transition model as the "policy" is interesting and could be useful to other researchers. The empirical results, particularly in Figures 3 and 7, seem to show that P2P can potentially reduce model error over multi-step rollouts.

The primary concern I have with this paper is its treatment of regions in state-action space with high model uncertainty. The authors seem to consider such regions as ones that should be avoided during reinforcement learning. On line 272, the authors state that their method instead "tends to generate trajectories heading to the regions with low uncertainty", while the baseline method MOPO "fails to prevent the trajectories from being absorbed into the highly uncertain region". This line of reasoning runs counter to the exploration-exploitation principle in reinforcement learning. Indeed, while the scenario in Figures 1 and 6 seems to have the goal in a region of low model uncertainty, it is certainly possible to encounter scenarios where the goal is instead in a region of high model uncertainty. Thus, exploration into regions with high model uncertainty in the online setting perhaps should be encouraged rather than avoided (subject to, e.g., safety constraints). This is a potentially critical limitation of the proposed work that is not discussed, and perhaps should be investigated with an additional experiment where the goal is located in a region of high model uncertainty.

---

> ### Author Response · Authors · 2022-08-02
> **Response to Reviewer q2Vv**
>
> We thank the reviewer for all the valuable comments. Point-by-point responses are provided below.
>
> **Q1: "Is the reward function** $R(s,a)$ **assumed to be known? If not, where and how do you learn** $R(s,a)$ **in Algorithm 1?"**
>
> **A1:**  $R(s, a)$ is assumed to be known in our analysis. We are sorry for confusion and have fixed this problem in the revised version (Section 2, Page 3, Line 103). Note that this is a commonly used assumption since the sample complexity of learning the reward function with supervised learning is a lower order term compared to the one of learning the transition model [1].
>
>
>
> **Q2: Clarification of** $R^m$.
>
> **A2:** **1) P2P-MPC:** During each training iteration, we first trains the model via traditional one-step prediction loss, and then trains the $\hat{R}^m$ network by taking transitions sampled from the environment dataset as input, and the prediction errors on these transitions as label. The prediction error of an environment transition $(s, a, r, s')$ are computed via $\\|\hat{s}'-s'\\|+\\|\hat{r}-r\\|$, where $\hat{s}', \hat{r}$ are sampled from $\hat{P}(\cdot, \cdot|s, a)$.
>
> **2) P2P-RL:** As mentioned in Section 3.3 in the paper, unlike P2P-MPC, P2P-RL does not actually generate the trajectories by interacting $\hat{P}$ with $\pi$. Instead, P2P-RL trains the model on the environment dataset and treats the model learning process as an offline RL problem, as the "decision maker" of the environment dataset is the true dynamics. Thus, regarding a transition $(s, a, r, s')$, $R^m$ can be directly approximated by computing $- \\|\hat{s}'-s'\\|-\\|\hat{r}-r\\|$ , where $\hat{s}', \hat{r}\sim\hat{P}(\cdot, \cdot|s, a)$.
>
>
>
> The above clarification has been added to Appendix B.3 (Appendix Page 3, Lines 38-48) in the revised version.
>
>
>
> **Q3: Clarification of the evaluations in Figure 4.**
>
> **A3:** The yellow curves represent the performance of P2P-MPC, which minimizes the multi-step loss on the trajectories generated by active interactions between the model and the current policy. The blue curves show the results of an ablation version of MBPO where the original one-step loss is replaced by a multi-step loss computed over the trajectories sampled from the environment dataset. The lengths of these trajectories are set to the same in this comparison. This clarification have been added to the revised version (Section 4.3, Page 8, Figure 4).
>
> **Q4: "In the online setting, could P2P underperform or fail in scenarios where the goal is in a region of high model uncertainty?"**
>
> **A4:** According to the reviewer's suggestion, we conducted a new experiment to investigate the case when the goal is in an uncertain region in the online setting. For the convenience of implementation, here the term "uncertainty" is equated with the epistemic uncertainty [2], which can be quantified as the amount of relevant real-world data. Therefore, a region with more data is considered to have lower uncertainty. Since in pure online settings the uncertainty of regions is hard to control during the training iterations, we first pretrain the model with an offline dataset and then switch to online training. The goal is allocated to the grey region where the relevant offline samples are partially discarded. The percent of discarded samples is set to 25%, 50%, 75% and 100% respectively and the results are given as follows:
>
>
> |         |       25%       |      50%       |      75%       |      100%      |
> | :-----: | :-------------: | :------------: | :------------: | :------------: |
> | P2P-MPC | $148.9\pm 35.9$ | $75.4\pm 31.6$ | $51.7\pm 29.8$ | $43.2\pm 25.1$ |
> |  MBPO   | $116.2\pm 35.6$ | $61.1\pm 34.8$ | $47.5\pm 35.1$ | $44.7\pm 30.2$ |
>
> As the degree of uncertainty increases, the performances of both methods degrade rapidly, but P2P-MPC still outperforms MBPO in all these cases except for the 100% case, where P2P-MPC achieves slightly worse performance in average but better stability with lower standard deviation. To give a possible explanation of these results, it is worth noting that **1)** P2P does not directly intervene the learning of policy or value function, but only improves the accuracy of the generated samples. As a result, the value function can still predict high value for uncertain regions and thus encourage the policy to explore them in the real environment; and **2)** in contrast, even if the goal is in a region with high uncertainty and the model does not prevent the policy from exploring this region in the model, the value function can still predict low value of this region due to the lack of relevant data and thus mislead the learning of policy.
>
> We have added these new results and explanations in Appendix D (Appendix Page 4, Lines 53-74).
>
>
> [1] Azar et al. Minimax pac bounds on the sample complexity of reinforcement learning with a generative model. 2013.
>
> [2] Chua et al. Deep Reinforcement Learning in a Handful of Trials using Probabilistic Dynamics Models. 2018.

---

> > ### Author Response · Authors · 2022-08-02
> > **Response to Reviewer q2Vv**
> >
> > **Q5: "Does generating trajectories heading to regions with low uncertainty run counter to the exploration-exploitation principle in reinforcement learning?"**
> >
> > **A5:** Generally speaking, the exploration-exploitation trade-off in RL mainly works on the real environment instead of the approximate model. Since it is hard for the uncertain regions to reflect the real dynamics accurately, exploring these regions can not only provides the agent with false information about the real environment, but also be prone to the *model exploitation* issue which severely hurts the asymptotic performance of the policy. From the theoretical perspective, preventing the policy from going into highly uncertain regions can reduce the accumulative model error, and thus guarantee a tighter performance lower bound and better policy improvement according to Theorem 1. Furthermore, note that P2P does not directly intervene the learning of value function or policy, hence the value function can still predict high value for uncertain regions and encourage the policy to explore them in the real environment. Overall, the focus of P2P is to learn a model which can quickly adapt to the current policy, so as to provide multi-step samples that are as accurate as possible for policy learning.

---

> > > ### Comment · Reviewer_q2Vv · 2022-08-09
> > > **Thanks for your responses!**
> > >
> > > I would like to thank the authors for their detailed responses, which corroborate my positive assessment of this paper.

---

> ### Author Response · Authors · 2022-08-05
> **Response to Reviewer q2Vv**
>
> Regarding the undefined notations, we have added clarifications in our revised version. (Page 2, Line 48 for the term "value", Page 5, Lines 149-150 for $R_{max}$ and $D_{TV}$)
>
> As for the use of the terms "policy-dynamics" and "model-policy", we have rewrote the relevant part to make it clearer to the readers in our revised version. (Page 6, 186-211)
>
> Thanks for your valuable suggestions, and please let us know if there are still confusions.

---

### Official Review · Reviewer_bd9p · 2022-07-12

**Rating:** 6
**Confidence:** 4
**Soundness:** 3 good
**Presentation:** 2 fair
**Contribution:** 2 fair

**Summary:**

The authors propose an MBRL framework named Plan to Predict (P2P), which treats the model rollout process as a sequential decision-making problem. The model in P2P can minimize the multi-step accumulative errors on the induced trajectories and thus quickly adapt to the state distribution of the current policy, the authors give a theoretical guarantee for P2P. Empirical results on several MuJoCo benchmark tasks verify the effectiveness of P2P.

**Questions:**


1. The description in Section 3.3 is a bit confusing. For P2P-MPC, Line 191 mentions "Besides, the reward R^m, which cannot be directly computed by definition, is approximated here using a neural network and trained on the environment dataset". How do we estimate R^m using a neural network, do you mean directly predicting the model error (sounds like a difficult job)? In P2P-RL, the authors mention (Line 205), "simply approximated by re-predicting the dynamics in the sampled transitions and computing the prediction errors for the current model-policy". It seems that R^m is obtained based on the data P^*, calculated by the \hat P prediction error. But if we use the model for multi-step rollout, in the successor time steps, we cannot get the ground-truthP^*(s’|\hat s,a), where \hat s is predicted by \hat P, how to calculate R^m? Also, for P2P-RL, a lot of technical details are mentioned here, such as: “updating the next state in each sampled transition by applying the current policy-dynamics“; ”adopting SAC with behavior cloning as the underlying learning algorithm“; ”DualDICE is also applied to correct the estimation of the state distribution“. I suggest that the authors can add pseudocode or a more complete description in the appendix to improve the readability of this part.


2. The author's explanation for the poor performance of the P2P-RL method (in Line 230) is that "we find that P2P-RL sometimes struggles to balance the loss of behavior cloning and RL, leading to the difficulty in hyperparameter tuning and the instability of learning network parameters”. But we can find that in the Hopper environment, the cumulative error of P2P-RL and P2P-MPC algorithms is similar (Figure 3), while the policy performance of P2P-RL is twice as bad. This seems to be contrary to the explanation proposed by the authors.

3. I think some related work is missing in this article. In recent years, a series of practical work and theoretical analyses have attempted to learn the model using the "treats the model rollout process as a sequential decision-making problem" framework. Its solution is similar but not exactly the same as P2P-RL. I recommend the author to read the following articles and add to the related work, which may inspire the author's follow-up research and improvement work [1,2,3].


[1] Wenjie Shang et al. Environment Reconstruction with Hidden Confounders for Reinforcement Learning based Recommendation.  2019.

[2] Tian Xu et al. Error Bounds of Imitating Policies and Environments. 2020.

[3] Xiong-Hui Chen et al. Adversarial Counterfactual Environment Model Learning. 2022.






**Limitations:**


Overall, the article is sound and has no major drawbacks to me. If the authors clarify the proposed questions and improve the writing of Section 3.3, I will consider increasing the score of the article.

**Strengths And Weaknesses:**

Strengths
1. The article is overall easy to follow except Section 3.3 (see Questions below);
2. The motivation to fix the mismatch between model learning and model usage is reasonable and valuable.
3. The experiment is well-designed to demonstrate the mechanism of P2P.

Weaknesses


The article is sound and has no major drawbacks to me. But there are several questions for the authors to further clarify (see below).

---

> ### Author Response · Authors · 2022-08-02
> **Response to Reviewer bd9p**
>
> We thank the reviewer for all these valuable comments. We provide point-by-point responses below.
>
> **Q1: About the implementation details.**
>
> **1) In P2P-MPC, "How do we estimate $R^m$ using a neural network, do you mean directly predicting the model error?"**
>
> **A1(1):** Yes, the reviewer has the right understanding, that is, the $\hat{R}^m$ network is trained to predict the model error. During each training iteration, P2P-MPC first trains the model via traditional one-step prediction loss, and then trains the $\hat{R}^m$ network by taking transitions sampled from the environment dataset as input, and the prediction errors on these transitions as label. The prediction error of an environment transition $(s, a, r, s')$ are computed via $\\|\hat{s}'-s'\\|+\\|\hat{r}-r\\|$, where $\hat{s}', \hat{r}$ are sampled from $\hat{P}(\cdot, \cdot|s, a)$.
>
> **2) In P2P-RL, "If we use the model for multi-step rollout, in the successor time steps, we cannot get the ground-truth** $P^*(s’|\hat s,a)$**, where $\hat{s}$ is predicted by $\hat{P}$, how to calculate** $R^m$**?"**
>
> **A1(2):** Unlike P2P-MPC, P2P-RL does not actually use the trajectories generated by the interaction of $\hat{P}$ and $\pi$. Instead, P2P-RL trains the model on the environment dataset and treats the model learning process as an offline RL problem, where the true dynamics becomes the "decision maker" of the environment dataset in our problem formulation. Thus, regarding a transition $(s, a, r, s')$, $R^m$ can be directly approximated by computing $-\\|\hat{s}'-s'\\|-\\|\hat{r}-r\\|$ , where $\hat{s}', \hat{r}$ are sampled from $\hat{P}(\cdot, \cdot|s, a)$.
>
>
>
> Regarding the mentioned details above and the rest of the concerns about P2P-RL, we have taken the reviewer's suggestion and added a more complete description as well as pseudocodes in Appendix B.3 (Appendix Page 3, Lines 38-48) and Appendix E (Appendix Page 4, Lines 75-77) to further improve the readability.
>
> **Q2: Explanation of the contradiction between the cumulative error and the performance of P2P-RL and P2P-MPC in Hopper.**
>
> **A2:** In our problem formulation, the final policy performance is not only determined by the accumulative model error but also affected by other factors (e.g., the value function approximation in the model learning phase of P2P-RL), which may result in the contradiction mentioned by the reviewer.
>
>
>
> **Q3: About the related work.**
>
> **A3:** Thanks for recommending these related articles.
>
> - Shang et al. [1] propose an environment reconstruction method which models the influence of the hidden confounder on the environment by treating the platform, the user and the confounder as three agents interacting with each other. They focus on the offline setting (i.e., RL-based recommendation) and simultaneously train the model and the policy using a multi-agent imitation learning method.
> - Xu et al. [2] treat the model as a dual agent and analyze the error bounds of the model. They propose to train the model using imitation learning methods. Chen et al. [3] also consider multi-step model error, but mainly focus on handling counterfactual data queried by adversarial policies. Unlike the above two work that both focus solely on model learning, P2P aims at proposing a general MBRL framework and attempts to adapt the model to the continuously updating policy quickly.
>
> While the above work shares some similarity with our work, the targeted problems and the proposed solutions are not exactly the same as P2P. We have added these references into the Related Work section in the revised version (Appendix F, Page 5, Lines 114-125).
>
> [1] Wenjie Shang et al. Environment Reconstruction with Hidden Confounders for Reinforcement Learning based Recommendation. 2019.
>
> [2] Tian Xu et al. Error Bounds of Imitating Policies and Environments. 2020.
>
> [3] Xiong-Hui Chen et al. Adversarial Counterfactual Environment Model Learning. 2022.

---

> > ### Comment · Reviewer_bd9p · 2022-08-06
> > **Response**
> >
> > Thanks for the detailed response. The response solves my several concerns. I have one further question on P2P-MPC. I found that the model learning process (Line 3-4 in Alg. 1)  is in the loop of ``N epochs'' but the current policy \pi does not affect the training process of model learning. If I understand correctly,  we can move it out as a pre-trained process (out of the loop of Line 2-20), right?

---

> > > ### Author Response · Authors · 2022-08-06
> > > **Response to Reviewer bd9p**
> > >
> > > We think that the reviewer may refer to the previous version (uploaded before the author response deadline) where the pseudo codes have some mistakes in terms of the loops. We are very sorry for this confusion.
> > >
> > > Please refer to the latest revision of our paper (uploaded yesterday) where Lines 6-20 are all in the loop of ``N epochs''.

---

> > > > ### Comment · Reviewer_bd9p · 2022-08-06
> > > > **Response**
> > > >
> > > > I am sure that I am using the latest version. In Line 3-4 in Alg. 1 (P2P-MPC),  we train $\hat P$ and $\hat R$ just based on the static offline dataset $\mathcal{D}_e$. So I think that obviously we can just pre-train $\hat P$ and $\hat R$ and fix them for $\pi$ learning (Line 5-18), instead of learning $\pi$ with imperfect $\hat P$ and $\hat R$. Are these some special designs or did I miss something?

---

> > > > > ### Author Response · Authors · 2022-08-07
> > > > > **Response to Reviewer bd9p**
> > > > >
> > > > > Yes, the reviewer is right, in the offline setting we pre-train $\hat{P}$ and $\hat{R}$ and fix them for policy learning. But please note that the pseudo codes we provide are for the online setting, where incremental environment data is collected and added to $\mathcal{D}_e$ in each iteration (Line 6). We will make this clearer in the Algorithm caption.

---

### Meta-Review · Area_Chair_NLg8 · 2022-08-29

**Recommendation:** Accept
**Confidence:** Certain

**Metareview:**

All the reviewers agree that this is a good paper. The idea is original and the paper has good empirical results. There were some confusions, which were resolved during the discussions and the revised paper. I recommend this paper to be accepted, possibly as a spotlight presentation.

I enlist a few concerns below, so that the authors can improve their paper. Some of them are by the reviewers, and some of them are by myself.

- Be more clear about how R^m is estimated. This is discussed in Appendix B.3 of the revised version, but given its importance to the algorithm, the authors may want to consider discussing it in the main body.

- Reviewer q2Vv mentioned their concerns about preventing the agent to go to the uncertain regions, which may prevent the exploration. The authors answered that "the exploration-exploitation tradeoff in RL mainly works on real environments instead of the approximate models".
This is not entirely accurate. Methods based on the optimism in face of uncertainty, such as UCRL, actually try to exploit the uncertainty of the model. If the model's promise of large return turns out to be false, due to its large uncertainty, we have gathered useful information and decreased our uncertainty.

- I feel that there is a gap between the theoretical results and the algorithm. It does not seem that the optimizer of the the model MDP (Definition 1), which is optimized on L5 of Algorithm 1, is the same as the optimizer of the upper bound on Theorem 1, used to justify the algorithm.
For example, the $e^m_t$ term is based on maximum over actions of the TV error between the model and the true environment, weighted according to state distribution induced by the model. On the other hand, $R^m$ in the model MDP (after taking the expectation over $s_{t+1}$ coming from distribution $P^m$), seems to be the chi-squared divergence, which is also weighted by the policy $\pi$. They are not the same.
It is OK if they are slightly different for practical purposes, as long as one can show their relation and be clear about it.

- The inequality at the beginning of Section 3.2 (Theoretical Results) requires $|J^\hat{P}(\pi) - J(\pi)|$ be smaller than C for both the new policy pi and the old policy $\pi_D$.
Disregarding the previous issue (or assuming that it can be resolved), solving the optimization problem defined on L5 of Algorithm 1 only guarantees $|J^\hat{P}(\pi) - J(\pi)|$ to be small for the current policy $\pi_D$, and not the optimized one. As such, the inequality is not satisfied, even if the algorithm works well.
Am I missing something? Please clarify it in the revised paper.

- It is claimed on L154 that monotonic policy improvement can be achieved by solving the update rule (2). I don't think it is correct. We need the value of the maximizer to be larger than $J(\pi_D)$, which may not always be the case.

**Award:**

No

---

### Decision · Program_Chairs · 2022-09-14

Accept